# *Staphylococcus aureus* Genomic Analysis and Outcomes in Patients with Bone and Joint Infections: A Systematic Review

**DOI:** 10.3390/ijms24043234

**Published:** 2023-02-06

**Authors:** Kevin Bouiller, Michael Z. David

**Affiliations:** 1Division of Infectious Diseases, Department of Medicine, University of Pennsylvania, Philadelphia, PA 19104, USA; 2Department of Infectious Diseases, Centre Hospitalier Universitaire CHU Besançon, F-25000 Besançon, France; 3UMR-CNRS 6249 Chrono-Environnement, Université de Franche-Comté, F-25000 Besançon, France

**Keywords:** *Staphylococcus aureus*, bone and joint infections, prosthetic joint infection, whole genome sequencing, outcome, osteomyelitis, septic arthritis, Panton Valentine leucocidin, *agr*

## Abstract

Many studies have been published assessing the association between the presence of *S. aureus* genes and outcomes in patients with bone and joint infections (BJI), but it is not known if they have had similar findings. A systematic literature review was performed. All available data on studies in Pubmed between January 2000 to October 2022 reporting the genetic characteristics of *S. aureus* and the outcomes of BJIs were analyzed. BJI included prosthetic joint infection (PJI), osteomyelitis (OM), diabetic foot infection (DFI), and septic arthritis. Because of the heterogeneity of studies and outcomes, no meta-analysis was performed. With the search strategy, 34 articles were included: 15 articles on children and 19 articles on adults. In children, most BJI studied were OM (*n* = 13) and septic arthritis (*n =* 9). Panton Valentine leucocidin (PVL) genes were associated with higher biological inflammatory markers at presentation (*n =* 4 studies), more febrile days (*n =* 3), and more complicated/severe infection (*n =* 4). Other genes were reported anecdotally associated with poor outcomes. In adults, six studies reported outcomes in patients with PJI, 2 with DFI, 3 with OM, and 3 with various BJI. Several genes were associated with a variety of poor outcomes in adults, but studies found contradictory results. Whereas PVL genes were associated with poor outcomes in children, no specific genes were reported similarly in adults. Additional studies with homogenous BJI and larger sample sizes are needed.

## 1. Introduction

*Staphylococcus aureus (S. aureus*) is the most common pathogen in almost all types of bone and joint infections (BJI), mainly due to its ability to adhere and to invade host tissues, evade immune defenses, and form biofilm in cases of prosthetic joint infection (PJI) or foreign-body associated osteomyelitis (OM) [1]. Outcomes of *S. aureus* BJI are closely related to host criteria (body mass index, tobacco, comorbidities, immunosuppression), medical/surgical management (antimicrobial therapy, removal of foreign bodies, debridement and irrigation), and phenotypic characteristics of the bacteria (resistance to methicillin, rifampicin or fluroquinolones) [1,2,3]. However, the role of the genetic background of the bacteria on the outcomes remains unknown. Researchers have associated both clinical characteristics and outcomes of the *S. aureus* bloodstream infection with different *S. aureus* clonal complexes (CCs) or production of toxins (Panton Valentine leucocidin, PVL) [4,5,6]. However, it is not known if, in cases of BJI, there are similar associations. In a recent literature review aiming to address the distribution of *S. aureus* CCs in outcomes of BJI, all major *S. aureus* clones appear to be capable of causing BJI, and no specific clone was related to BJI [7]. However, no additional genetic analysis (such as presence of virulence factors) and/or outcome was analyzed in the review. The aim of this systematic review was to evaluate *S. aureus* genes and their association with outcomes in adults and children with BJI. We included studies that compared *S. aureus* strains from first and recurrent infection episodes.

## 2. Materials and Methods

### 2.1. Literature Searched

A systematic literature review was conducted according to the Preferred Reporting Items for Systematic Review and Meta-Analyses (PRISMA) statement. The literature search was performed in Pubmed for studies published between January 2000 to October 2022. We searched for all citations on BJI including *S. aureus* gene analysis and citations that reported the index and recurrent strain type in patients with recurrent BJI. BJIs included PJI, OM (with or without a foreign body), diabetic foot infection (DFI), and septic arthritis. The search strategy is available in the Appendix A. Studies in all languages were analyzed. Studies were excluded if they were (1) studies lacking evaluation of any genes with outcomes, (2) general reviews, (3) case reports, except for case reports with relapses or recurrent BJI and genetic analyses of the initial and one or more recurrent or relapsing strains, (4) laboratory animal or in vitro studies, (5) diagnostic assay evaluation, (6) studies of bacteria other than *S. aureus*, and (7) human genetic studies. All studies were screened by K.B. and reviewed by M.Z.D. Uncertainties about the inclusion of studies were resolved by discussion. The review and protocol were not registered.

### 2.2. Data Extraction

The following data were extracted from each selected study: year of publication, country, population description (adults, children), study design, number of isolates, resistance to methicillin, type of genetic analysis (PCR, WGS, DNA microarray, PFGE), type of BJI (OM, PJI, septic arthritis), and outcomes. Outcome was defined as severity of infection (biological serum markers, duration of fever, sepsis, septic shock, ICU admission, mortality), and/or complication (bone fracture, abscess, etc.), and/or relapse, and/or recurrent infection, and/or any relevant outcomes defined in the study. In case reports, data on the genetic differences between the first strain and any relapse or recurrent strains were recorded.

### 2.3. Data Synthesis and Analysis

Because of the small number of studies and the heterogeneity of outcomes in included studies, no meta-analysis was performed. Because the included studies were neither randomized controlled trials nor, in some cases, comparative studies, traditional methods for assessment of risk of bias were not applicable. We collected and tabulated data on methodologic variables that we deemed relevant for the review: study design, age group (children and adults), number of participants, methicillin sensitivity, type of BJI, and genes analyzed.

## 3. Results and Discussion

### 3.1. Studies Included

Figure 1 shows the flow chart of the screening process of our literature review. With the search strategy, 503 articles were identified in Pubmed. After abstract screening, 111 (22%) were assessed for eligibility with full text analysis. One article appeared to meet the inclusion criteria, but the number of genes analyzed in each group and the statistical analysis were not reported [8]. Ultimately, 34 articles were included in the review, 15 articles in children, and 19 articles in adult patients (see Appendix A). Among the 34 articles, six case reports were included, all in adult patients.

### 3.2. BJI in Children

#### 3.2.1. Studies Included

We included 15 studies that reported BJI in children. Most BJI studied were either acute hematogenous OM (*n =* 13) or septic arthritis (*n =* 9). Three studies included only methicillin susceptible *S. aureus* (MSSA) isolates [9,10,11], and one study only methicillin resistant *S. aureus* (MRSA) isolates [12]. In other studies, the prevalence of MRSA varied from 2.7% (Morocco) to 88% (USA). Only two studies performed whole genome sequencing (WGS) analysis [12,13]. The 13 other studies used specific PCR assays to detect genes, specifically PVL (13/13), *mecA* (3/13), the *agr* system (4/13), or other genes (3/13) (Table 1).

#### 3.2.2. PVL

Prevalence of PVL varied from 17% to 25% in studies with MSSA isolates, and 40 to 100% in studies with MRSA isolates. Four studies reported an association between more elevated biological inflammatory markers (ESR, C-reactive protein, or white blood cell count) and presence of PVL [10,15,17,23], whereas one study did not report such differences [22]. Two studies reported more febrile days in children with BJI with a PVL-positive isolate infection (days, mean ± SD: 4.2 ± 3.6 vs. 2.1 ± 2.5 [*p* = 0.017] and 28.6 ± 23.2 vs. 2.8 ± 2.2 [*p* < 0.001]) [20,21]. Six studies reported more severe infections (ICU hospitalization, death) or complicated infections (fracture, deep venous thrombosis, abscess) in children with PVL-positive strains in univariate analysis [9,15,17,18,19,20]. However, among three studies with multivariate analysis only one showed that PVL-positive strains were associated with severe infection (ICU hospitalization or death) [19].

Four studies evaluated the relationship between PVL-positive strains and complications in BJI in long term follow-up. One study with a mean time of follow-up of 25 months found that 12/14 children with PVL-positive strains had complications (including 2 late fractures, 2 leg-length discrepancies, and 4 cases of radiographic bone abnormalities) versus none of the 17 children with PVL-negative strains [18]. Belthur et al. compared patients with or without fracture with a mean follow-up of 22.4 months (range, 6.5–41.5) in the fracture group versus 10.4 months (range 3–34) in the non-fracture group. The mean time from disease onset to fracture was 72.1 days (range, 20–150 days). Frequency of PVL-positive strains was not significantly different in the 2 groups (100% vs. 92%, *p* = 0.565) [14]. Martinez-Aguilar et al. reported 3/33 chronic OM on follow-up in patients with PVL-positive strains versus no complication in children with PVL-negative strains. The mean follow-up was 20 ± 1 months [20]. In the study of McNeil et al. 13 (5.2%), infections progressed to chronic OM with a median follow-up of 92.5 days (interquartile range 37–324). However, PVL strains were not associated with progression to chronic OM [11]

#### 3.2.3. Sequence Type/Clonal Complex

One study found that the USA300-0114 clone was associated with bone fracture [14]. Another study found patients infected with the USA300 clone had a more severe inflammatory syndrome than with other clones [10]. In another study, the USA300 clone and PVL-positive strains were associated with ICU admission in univariate analysis but not in multivariate analysis [9].

#### 3.2.4. Other Genes

The *fnbB* gene was evaluated in three studies (one using specific PCR, two using WGS), but no significant difference associated with outcomes was reported [15,16,20].

Two studies with PCR detection of *clfA*, *cna*, *fib*, *clfB*, *bbp*, *eno*, and *ebpS* did not find any differences between outcomes and the presence of these genes in children with BJI [16,20].

In two studies, *agr* type III was associated with orthopedic complications and chronic OM in multivariate analysis [11,21].

In the two studies with WGS analysis, one found 40 genes associated with increased severity of illness, including the presence of *lukF-PV* genes [13]. The second study did not show any association between virulence genes and severity [12].

#### 3.2.5. Quality of Studies

Of 15 included studies, four were prospective, but only one was multicenter. The inclusion criteria were homogenous; however, some studies included only AHO, septic arthritis, or both. Sample size was small in most studies, and multivariate analysis was possible in only 3 of them. The outcomes were well defined but differed among studies: two studies used the Modified Severity of Illness scoring system, whereas other studies used a composite score (ICU and/or death) or ICU admission alone. Complications were also defined differently in these studies. Only two of the studies used WGS analysis, but one study had only 12 patients which limited power to compare groups. In the other study, no multivariate analysis was performed, probably due to the high number of variables with significant differences on univariate analysis. No studies reported an analysis for an outcome of relapse or recurrent infection.

### 3.3. BJI in Adults

#### 3.3.1. Studies Included

We included 14 studies that compared the presence of genes and outcomes in BJI. Six studies reported outcomes in patients with PJI, two in patients with DFI, three with OM, and three with various BJI. MSSA isolates were the only bacteria included in three studies and MRSA in two studies. In other studies, the prevalence of MRSA varied from 1% (Sweden) to 70% (India). Detection of genes was performed by a multiplex PCR in one study, DNA microarrays in three studies, and an analysis of WGS data in three studies. The other seven studies used PCR targeting PVL (*n =* 3), clonal complex (*n =* 3), and a number of different genes (PVL, *mecA*, *tsst*, *eta*, *etb*) (*n =* 1) (Table 2).

#### 3.3.2. PVL

Among 14 studies included, 11 reported analyses of PVL genes and their outcomes; three studies included patients with OM, six with PJI, one with DFI, and one with multiple types of BJI. Only one study identified an association between PVL-positive strains and chronic OM with a longer duration of evolution of OM (mean ± SD, 192.1 ± 150 months versus 50.0 ± 150 months [*p* = 0.00060]) [24].

#### 3.3.3. Sequence Type/Clonal Complex

Two studies identified differences comparing CC398 strains vs. other strains with a clinical outcome, but there were conflicting results in the studies. In one study, patients with *S. aureus* CC398 DFI were compared with patients infected with other *S. aureus* clones. Patients with *S. aureus* CC398 DFI had a more severe infection (International Working Group of the Diabetic Foot-Infectious Diseases Society of America classification grade 4) compared to other clones [25]. In the other study, the authors compared patients with *S. aureus* CC398 with BJI to patients with BJI caused by other *S. aureus* clones. Patients with *S. aureus* CC398 had a lower treatment failure rate than with other clones (0% vs. 37.3% [*p* = 0.032]) [26]. In a study of patients with acute PJI, the clone ST239 was associated with a higher number of surgical debridements (ST239 *n =* 10/11 vs. ST8 *n =* 4/10, *p* < 0.05), a higher pre-operative CRP level (mean ± SD, 279.16 ± 109.53 mg/L vs. 66.8 ± 57.56 mg/L, *p* < 0.05) and having fevers more often than other clones [27]. In three other studies, no difference was found comparing CC or ST and their outcomes [28,29,30].

#### 3.3.4. Other Virulence Genes

In 2 studies that evaluated the presence of specific genes and the outcomes of chronic OM, results were discordant. In one study, *bbp* and *sei* genes were associated with chronic OM, whereas no tested gene (*bbp*, *cna*, *fnbB*, *sdrD*, *sdrE*) was associated with this outcome in the other study [24,31].

In one study, the genes *sak*, *scn,* and *chp* had a higher frequency in patients with the failure of PJI treatment, but this was not statistically significant (*sak* 80.0% vs. 37.5% [*p* = 0.145]; *chp* 80.0% vs. 25.0% [*p* = 0.054]; *scn* 80.0% vs. 37.5% [*p* = 0.145]) [32].

In one study with WGS analysis, the serine protease gene *splE* and resistance gene *blaZ* were significantly less frequent in patients with resolved *S. aureus* PJI than unresolved infections (*splE* 40% vs. 70% [*p* ≤ 0.05]; *blaZ* 50% vs. 90% [*p* ≤ 0.01]) [29].

In the second WGS study, *cap5H*, *cap5J*, and *cap5K* genes were associated with treatment failure in patients with PJI, whereas *cap8H*, *cap8I*, *cap8J*, *cap8K*, and *sspP* were associated with treatment success, and *cap8H* and *cap8K* with the eradication of *S. aureus* [30].

The third study, using the analysis of WGS data did not show any differences between genome analysis and outcomes in patients with PJI [33].

In one study with PJI, infection by strains that were *agr* type II showed a trend toward association with treatment failure (46.2% vs. 24.1%, *p* = 0.099) [28], whereas *agr* type II was associated with resolution of infection (*n =* 8/10, 80% vs. *n =* 2/10, 20% [*p* = 0.0256]) in another study [29].

**Table 2 ijms-24-03234-t002:** Association between genes and outcome in bone and joint infections in adults.

Reference	Study Design	Country	Years of Study	Number of Patients/Strains	MRSA n, (%)	Type of BJI	Genomic Analysis	Comparison	Follow-Up	Genotype and Outcome
Banerjee et al., 2020 [34]	Prospective single-center	India	2018–2019	60/60	40/60 (67)	OM, arthritis, spondylodiscitis Presence of foreign bodies (*n =* 30)	PCR *lukS-PV*, *lukF-PV*, *lukD* and *lukE*	Recovered (*n =* 46) vs. not recovered (*n =* 14)	All patients were followed up for six months after the end of treatment for the initial infection	In BJI with foreign body (*n =* 30), factors associated with unresolved infection (univariate analysis): Age: OR (95% CI) = 0.94 (0.88–1.0), *p* = 0.049 Number of surgical interventions: OR (95% CI) = 10.68 (1.37–83.52), *p* = 0.024 Duration of antibiotics: OR (95% CI) = 1.07 (1.01–1.13), *p* = 0.029 Presence of *luk* genes was not associated with unresolved infection: OR (95% CI) = 2.0 (0.38–10.51), *p* = 0.413 In BJI without foreign body (*n =* 30), no factor significantly differed Presence of any *luk* genes: OR (95% CI) = 0.44 (0.22-23.27), p = 0.413
Bouiller et al., 2020 [25]	Retrospective single-center	France	2010–2017	244/244	37/244 (15)	DFI with OM	PCR CC398-lineage specific	CC398 (*n =* 37) vs. non-CC398 (*n =* 207)	ND	In multivariate analysis, a patient with *S. aureus* CC398 DFI had more severe infection (IWGDF-IDSA grade 4): 23/37, 62% vs. 33/207, 16%, OR (95% CI) = 8.5 (3.5–20.7), *p* < 0.001
Chen et al., 2022 [27]	Retrospective single-center	Taiwan	2016–2019	36/36	0	Acute PJI	PCR *lukS-PV*, *lukF-PV*, MLST	ST239 (*n =* 11) vs. ST8 (*n =* 10) vs. ST59 (*n =* 8) vs. other ST (*n =* 7)	ND	Number of surgical debridements ST239 vs. ST8: *n =* 10/11, 91% vs. *n =* 4/10, 40%, *p* < 0.05 ST239 vs. ST59: *n =* 10/11, 91% vs. *n =* 6/8, 75%, *p* > 0.05 ST239 vs. other ST: *n =* 10/11, 91% vs. *n =* 7/7, 100%, *p* > 0.05 Number of hospital admissions ST239 vs. ST8: *n =* 7/11, 64% vs. *n =* 5/10 50%, *p* > 0.05 ST239 vs. ST59: *n =* 7/11, 64% vs. *n =* 5/8 63%, *p* > 0.05 ST239 vs. other ST: *n =* 7/11, 64% vs. *n =* 6/7, 86%, *p* > 0.05 Pre-operative CRP (mg/L, mean ± SD): ST239 vs. ST8: 279.16 ± 109.53 vs. 66.8 ± 57.56, *p* < 0.05 ST239 vs. ST59: 279.16 ± 109.53 vs. 184.29 ± 90.60, *p* < 0.05 ST239 vs. other ST: 279.16 ± 109.53 vs. 118.19 ± 100, *p* < 0.05 Fever (no p-value provided) ST239 vs. ST8: *n =* 7/11, 64% vs. *n =* 1/10, 10% ST239 vs. ST59: *n =* 7/11, 64% vs. *n =* 0/8 ST239 vs. other ST: *n =* 7/11, 64% vs. *n =* 1/7, 14% Prevalence of PVL was higher in ST8 than other STs: ST8: *n =* 9/10 (90%) ST59: *n =* 6/8 (75%) ST239: *n =* 5/11 (46%)
Jiang et al., 2017 [24]	Retrospective, single-center	China	2013–2015	60/60	9/60 (15)	OM	Multiplex PCR ^1^	PVL+ (*n =* 17) vs. PVL− (*n =* 43) Duration time of OM < 24 months or >24 months and <20 years or >20 years	ND	OM duration was significantly longer in PVL-positive patients (mean ± SD): 192.1 ± 150 mo versus 50.0 ± 150 mo, *p* = 0.00060 PVL gene presence was positively associated both with OM duration of greater than 24 months (*p* = 0.00030) and greater than 20 years (*p* = 0.00030) Presence of *bbp* (*p* = 0.016) and *sei* (*p* = 0.041) genes were associated with OM duration of longer than 20 years
Kalinka et al., 2014 [31]	Retrospective multicenter	Argentina	ND	21/21	7/21 (33)	OM	PCR and/or RT-PCR ^2^	Acute OM (<2 months) (*n =* 11) vs. chronic OM (>2 months) (*n =* 10)	ND	Little difference between the presence or absence of genes and acute or chronic OM, but no statistical analysis was performed to compare the 2 groups *bbp*: 9% vs. 0% *cna*: 27% vs. 10% *fnbB*: 82% vs. 100% *sdrD*: 100% vs. 100% *sdrE*: 82% vs. 100%
Muñoz-Gallego et al., 2017 [32]	Retrospective, single-center	Spain	2005–2015	18/18	18/18 (100)	PJI	DNA microarrays ^3^ PFGE, MLST	Failure ^4^ vs. cure	688 days (median)	No significant differences but a trend towards a higher frequency in failure group of: *sak:* 80.0% vs. 37.5% (*p* = 0.145) *chp:* 80.0% vs. 25.0% (*p* = 0.054) *scn:* 80.0% vs. 37.5% (*p* = 0.145)
Muñoz-Gallego et al., 2020 [28]	Prospective multicenter	Spain	2016–2017	88/88	20/88 (22.7)	PJI	DNA microarrays ^3^	Failure ^5^ vs. cure	1 year	No genotypic or phenotypic characteristics predicted failure, except vancomycin MIC ≥ 1.5 mg/L (23.1% failure vs. 3.4% cure, *p* = 0.044). Clonal complex (*p* = 0.574), and strains with different *agr* type (*p* = 0.434) were not predictive of failure Trend toward greater presence of *S. aureus* belonging to CC5 (30.8% vs. 13.8%, *p* = 0.192) and *agr* type II (46.2% vs. 24.1%, *p* = 0.099) among the failures
Muñoz-Gallego et al., 2022 [33]	Prospective multicenter	Spain	2016–2107	14/20	2/18 ^6^ (11)	PJI with DAIR (symptoms < 21 days)	WGS	Relapse/persistent (*n =* 6) vs. Cured (*n =* 8)	1 year	No association between the presence or absence of genes and persistent or relapsing PJI versus resolved PJI No significant difference between strains causing persistent or relapsing PJI and those with a favorable outcome
Peyrani et al., 2012 [35]	Retrospective, single-center	USA	2007–2008	50/50	50/50 (100)	OM	PFGE, PCR PVL gene (unspecified)	Success ^7^ (*n =* 37) vs. failure (*n =* 13)	>1 year	No correlation with USA300 pulsotype, PVL, SCC*mec* or *agr* type USA300: *n =* 21/37, 56.8% vs. *N =* 6/13, 46.2% (*p* = 0.537) PVL: *n =* 10/37, 27% vs. *N =* 2/13, 15.4% (*p* > 0.05) SCC*mec* type II: *n =* 16/37, 43.2% vs. *N =* 7/13, 53.9% (*p* > 0.05) *agr* type II: *n =* 12/37, 40% vs. *N =* 5/13, 55.6% (*p* > 0.05) The PVL gene was identified in 44% of USA300 isolates and in none of the non–USA300 isolates
Trobos et al., 2022 [29]	Retrospective, single-center	Sweden	2012–2015	38/45	3/45 (6.7)	PJI	WGS	Resolved ^8^ (*n =* 22) vs. unresolved infection (*n =* 23)	3.5–5 years	None of the CC types were associated with the outcome: CC15: *n =* 4/5, 80% vs. *n =* 1/5, 20% (*p* = 0.1399) CC30: *n =* 2/7, 28.6% vs. *N =* 5/7, 71.4% (*p* = 0.2163) CC45: *n =* 7/12, 58.3% vs. *N =* 5/12, 41.7% (*p* = 0.4447) CC8: *n =* ¼, 25% vs. *n =* ¾, 75% (*p* = 0.3167) *agr* type II strains were significantly associated with a resolved outcome *agr* type II: *n =* 8/10, 80% vs. *n =* 2/10, 20% (*p* = 0.0256) Only the serine protease gene *splE* and *blaZ* were significantly less frequent in *S. aureus* from resolved than unresolved infections: *splE*: *n =* 9/22, 40% vs. *n =* 16/23, 70% (*p* ≤ 0.05) *blaZ*: *n =* 11/22, 50% vs. *n =* 21/23, 90% (*p* ≤ 0.01)
Valour et al., 2014 [26]	Retrospective single-center	France	2011–2012	75/75	0	Native septic arthritis, *n =* 10 (13%) PJI, *n =* 16 (21.3%) OM, *n =* 38 (50.7%) Vertebral OM, *n =* 11 (14.7%)	PCR CC398-specific	CC398 (*n =* 8) vs. Other clones (*n =* 67)	ND	Less severe biological inflammatory syndrome ^9^ with CC398: *n =* 5/8, 62.5% vs. *n =* 62/67, 92.5% (*p* = 0.035) Lower treatment failure ^10^ rates with CC398: *n =* 0/8 vs. *n =* 25/67, 37.3% (*p* = 0.032)
Valour et al., 2015 [36]	Retrospective single-center	France	2001–2011	95/95	0	Arthritis, *n =* 12 (13%) Non-vertebral OM, *n =* 12 (13%) Vertebral OM, *n =* 7 (7%) Foreign body infection, *n =* 64 (67%) (PJI, *n =* 29, OM, *n =* 32, others, *n =* 3)	DNA microarray ^3^	Acute ^11^ (*n =* 64) vs. Chronic BJI (*n =* 31)		No significant difference in distribution of clonal complexes comparing acute and chronic BJI (*p* = 0.408) No identification of virulence genes associated with BJI chronicity: Multivariate analysis results *sdrD*: OR (95% CI) = 0.365 (0.128–1.038), *p* = 0.059 Internalization rate: OR (95% CI) = 1.000 (1.000–1.000), *p* = 0.244 Delta-toxin negative: OR (95% CI) = 0.397 (0.708–8.112), *p* = 0.160
Viquez-Molina 2018 [37]	Prospective single-center	Costa Rica	2004–2016	58	35/58 (60.3)	DFI (58/58) OM (41/58)	PCR *mecA*, *lukS-PV*, *lukF-PV*, *tsst*, *etA*, *etB*	Lower extremity amputation vs. no lower extremity amputation	ND	In patients with DFI, the only predictive factor associated with the outcome of lower extremity amputation was the presence of necrosis: OR (95% CI) = 6.6 (1.5–28.4), *p* = 0.01 *mecA+* vs. *mecA−* *n =* 13/35, 37% vs. *n =* 4/23, 17% (*p* = 0.10) PVL*+* vs. PVL*−* *n =* 0/4 vs. *n =* 17/54, 31.5% (*p* = 0.23) *tsst*+ vs. *tsst*− *n =* 2/3, 67% vs. *n =* 15/55, 27% (*p* = 0.20) In 41 patients with OM associated DFI, presence of *mecA* was not associated with amputation: *n =* 4/14, 28.6% vs. *n =* 12/27, 44.4% (*p* = 0.32) Patient with *mecA*+ had longer time to wound healing: *mecA+* vs. *mecA−* (days, median [Q1, Q3]): 104 (52, 132) vs. 74 (45, 80) (*p* = 0.04)
Wildeman et al., 2020 [30]	Retrospective multicenter	Sweden	2004–2016 2012–2016	100	1/100 (1)	PJI	WGS	Treatment failure ^12^ 49/99 (49%) or no microbiological eradication ^13^ 40/99 (40%)	26 months (median)	Genes associated with treatment failure (success vs. failure): *cap5H: n =* 7/50, 14% vs. *n =* 17/49, 34%, *p* = 0.02, OR (95% CI) = 2.83 (1.06–7.60) *cap5J: n =* 7/50, 14% vs. *n =* 16/49, 32%, *p* = 0.03, OR (95% CI) = 2.55 (0.95–6.87) *cap5K: n =* 7/50, 14% vs. *n =* 16/49, 32%, *p* = 0.03, OR (95% CI) = 2.55 (0.95–6.87) Genes associated with treatment success: *cap8H: n =* 23/49, 46% vs. *n =* 36/50, 72% *p* = 0.01, OR (95% CI) = 0.38 (0.17–0.87) *cap8I: n =* 31/49, 62% vs. *n =* 41/50, 82%, *p* = 0.02, OR (95% CI) = 0.38 (0.15–0.97) *cap8J: n =* 32/49, 64% vs. *n =* 42/50, 84%, *p* = 0.02, OR (95% CI) = 0.35 (0.13–0.95) *cap8K: n =* 32/49, 64% vs. *n =* 42/50, 84%, *p* = 0.02, OR (95% CI) = 0.32 (0.13–0.81) *sspP: n =* 44/49, 88% vs. *n =* 50/50, 100%, *p* = 0.03 (odds ratio not provided) Genes associated with eradication: *cap8H: n =* 40/59, 68*%* vs. *n =* 18/40, 45%, *p* = 0.02, OR (95% CI) = 0.39 (0.17–0.90) *cap8K: n =* 46/59, 78% vs. *n =* 24/40, 60%, *p* = 0.05, OR (95% CI) = 0.44 (0.18–1.08) ^14^

BJI: bone and joint infection; CC: clonal complex; CRP: C-reactive protein; DNA: deoxyribonucleic acid; DFI: diabetic foot infection; IWGDF-IDSA: International Working Group of the Diabetic Foot-Infectious Diseases Society of America classification; MIC: minimum inhibitory concentration; MLST: multilocus sequence typing; MRSA: methicillin-resistant Staphylococcus aureus; ND: no data; OM: osteomyelitis; OR: odds ratio; PCR: polymerase chain reaction; PFGE: pulsed-field gel electrophoresis; PJI: prosthetic joint infection; PVL: Panton Valentine leucocidin; ST: sequence type; WGS: whole genome sequencing. ^1^ Including 11 adhesion associated genes (bbp, ebps, can, eno, icaA, icaD, fnbA, fnbB, fib, clfA, and clfB) and 25 exotoxin-associated genes (*lukS-PV, lukF-PV, tst, eta, etb, edin, psmα, lukM, lukED, hla, hlb, hld, hlg, hlgv, sea, seb, sec, sed, see, secanseh, sei, sej, sem, sen*, and *seo*). ^2^ *bbp, clfA, canB, cna, eap, efb, emp, fnbA, fnbB, isdA, sdrC, sdrD, sdrE, agrA, aroE, aur, clfA, clfB, eap, amp, fnbA, fnbB, gyrB, psmα, saeR, sarA, sdrE, sigB*, and *spa*. ^3^ DNA microarrays (*S. aureus* Genotyping Kit 2.0. Alere, Jena, Germany). The array covers 300 different targets related to approximately 185 different genes and their allelic variants. For the complete list of the target genes see Monecke et al., FEMS Immunol Med Microbiol, 53(2), 2008:237–251. ^4^ Death related to the infection, relapse, or persistence of the infection. If implant retention, need for salvage therapy, such as supplementary debridement > 30 days after the first surgery, prosthesis removal, or supplementary course of antibiotics; if prosthesis removal (i.e., chronic infections), failure also included the absence of a new prosthesis replacement. ^5^ Death from any cause within 90 days after surgery, persistent or relapsing signs of staphylococcal infection, and/or the need for salvage therapy due to *S. aureus*, including antimicrobial suppressive therapy and unplanned surgeries (except for extra debridements in the first 30 days after the initial therapeutic surgery). ^6^ No data for two strains. ^7^ Clinical Success: (a) patients with no clinical or laboratory evidence of infection with all antibiotics discontinued; (b) patients with no clinical or laboratory evidence of infection but continuing oral antibiotics because of a high risk for relapse. Clinical Failure: (a) patients experiencing clinical and laboratory deterioration during antibiotic therapy that required surgical intervention; (b) patients experiencing clinical and laboratory deterioration after discontinuation of antibiotic therapy. ^8^ Absence of infection with clinical or microbiological criteria and no further treatment (surgery or antibiotics). ^9^ CRP level >10 mg/L at the time of diagnosis. ^10^ Treatment failure: (a) persistence of septic symptoms despite appropriate surgical and medical treatment; (b) relapse owing to isolation of the same MSSA strain after cessation of treatment; or (c) the need for a new surgery for sepsis more than 5 days after the initial surgery. ^11^Infection lasting < 4 weeks. ^12^ At least one of the following: (a) need for prosthesis removal (one- or two-stage exchange, resection arthroplasty, or amputation); (b) persistent clinical and laboratory signs of infection; (c) need for suppressive antibiotic treatment of any pathogen including *S. aureus*; or (d) death during antibiotic treatment when no other evident explanation was apparent. ^13^ All of the following: (a) no signs of infection and no antibiotic treatment directed at *S. aureus* (regardless of whether the primary intervention required prosthesis removal, i.e., one-stage, two-stage, or resection arthroplasty); (b) no *S. aureus*-associated relapse after completed primary intervention; and (c) the patient did not die during PJI treatment. ^14^ When adjusting for age and sex, the associations for *cap5H*, *8H*, *8I*, *8J*, *8K*, and treatment outcome remained significant. *cap8H* was still significantly associated with eradication.

#### 3.3.5. Quality of Studies

Five studies reported outcomes of severity of infection [25,27] or determinants of chronic infection [24,31,36], whereas nine other studies used different definitions of outcome for failure, recurrence, or success [26,28,29,30,32,33,34,35,37] (Table 2).

In six studies of only PJI [27,28,29,30,32,33], inclusion criteria varied, with one study including acute PJI (without precise definition) [27] and one study including only PJI with DAIR treatment [33]. In four other studies, classification of PJI varied and the rate of DAIR treatment ranged from 53% to 83% [28,29,30,32]. Antimicrobial therapy, specifically rifampicin use, was reported in only three studies [28,29,30], but in one study data on *S. aureus* and *S. epidermidis* were not distinguished [29]. In the six studies of PJI, the rate of failure varied from 36 to 56% [27,28,29,30,32,33]. Adjusting analysis on confounding factors was performed in only one study [30].

In studies of OM, surgical therapy differed among studies depending on the type of BJI included. In three studies with device-related OM [26,34,36], data on rifampicin used was reported in only one [34]. In three OM studies, analysis of acute/chronic OM were reported with different definitions of acute infection (i.e., <1 month, <2 months and <24 months) and chronic infection (i.e., ≥1 month, >12 months, and ≥24 months) [24,31,36].

In studies of DFI, one study analyzed severity of infection [25], and another defined failure as lower limb amputation [37]. However, in this last mentioned study, patients with OM and with non-complicated SSTIs were analyzed together [37]. Results of the specific group of DFI with OM were only reported for the presence or absence of *mecA*.

### 3.4. Comparison of Initial and Relapse Strains in BJI with Treatment Failure

#### 3.4.1. Studies Included

Six studies comparing initial and relapse *S. aureus* strains in BJI after treatment failure were included: 3 case reports (with one patient) [38,39,40], and 3 case series with 3, 4 and 14 patients, respectively [33,41,42] (Table 3). Three studies analyzed only patients with PJI [33,40,41] and one study included OM with foreign bodies in one patient and PJI in two patients [42]. Two other studies included OM and native septic arthritis [38,39]. First and persistent strains were analyzed in two studies [38,39], relapse in two studies [40,42], and recurrent and relapse strains in two studies [33,41]. Two studies analyzed specifically small colony variant (SCV) strains [38,40]. All six studies used WGS to compare strains.

#### 3.4.2. Analysis of Genetic Differences between First and Recurrent/Relapse Isolates

The first study assessed the mutations responsible for genetic characteristics of a vancomycin-intermediate *S. aureus* (VISA) SCV in a patient with septic arthritis during long term treatment. Between the first and last VISA SCV isolates, 13 genetic differences in SCC*mec*, *mprF*, *cls2*, *clpX*, and *fabF* were found. The mutation of *fabF* (encodes a fatty acid synthase) was probably responsible for the SCV phenotype of recurrent strains [38].

In the second study, the authors characterized two different ST398 MSSA isolates causing septic knee arthritis and lack of response to antimicrobial therapy. No difference in virulence or resistance genes was found between the first and recurrent strains. In core genome SNPome analysis, only one SNP differed between the two strains [39].

The third study assessed genetic differences between a first MSSA isolate and an SCV isolate in a case of relapsing PJI. The loss of a 42.5 kb prophage in the genome of the relapse-SCV strain (φSa2) and three deletions, including a non-truncating deletion within the *rpoB* gene, were associated with the SCV isolate [40].

The fourth study evaluated mutations differentiating both the first and a relapse *S. aureus* strain in one patient with PJI and both the first and a recurrent strain in two patients with PJI. Mutations in genes coding for proteins involved in fibronectin binding (*ebh*, *fnbA*, *clfA*, and *clfB*), which distinguished later PJI isolates from the first PJI isolate were found in all three patients. Additionally, in the patient with the relapse strain, mutations in lysostaphin, multidrug resistance, pheromone binding, and epimerase were associated with the relapse strain [41].

The fifth study assessed genetic differences between the first and persisting or relapsing MSSA strains in one patient with foreign body OM and two patients with PJI. Authors reported six, eight, and nine mutations between different paired strains but with no effect on the major regulatory systems that are known to control the expression of virulence in *S. aureus* [42].

In the sixth study, the authors evaluated genetic differences between first and relapse isolates in three patients with PJI and first and recurrent isolates in other three patients with PJI. In five of the six patients, specific mutations were identified: in *bbp*, *sdrD*, *clfA*, *fnbA*, and *fnbB* for two patients, in *fmt* for one patient, and a deletion in DNA-3-methyladenine glycosylase in one patient [33].

#### 3.4.3. Quality of Studies

The studies were heterogeneous, and, thus, no meta-analysis was possible. In one study, clinical metadata for compared isolates were not clearly reported between relapse or persistent strains. Moreover, antimicrobial and surgical therapy were not reported [41].

### 3.5. Discussion

This review examines published studies assessing the presence or absence of *S. aureus* genomic content and clinical outcomes in patients with BJI. Whereas the carriage of PVL genes was correlated with increased disease severity in children with BJI, adult studies were usually limited by small sample size.

#### 3.5.1. Role of *S. aureus* Lineage

Some *S. aureus* lineages have been associated with poor outcomes in patients with bacteremia [6]. The best example is the USA300 lineage, belonging to ST8/CC8-MRSA-IV, which rapidly emerged in the late 1990s to become the dominant community-acquired MRSA strain in North America by 2004 [43]. Selection and diffusion of this clone is likely related to the acquisition of various mobile genetic elements, specifically *SCCmec* type IVa carrying the *mecA* gene, phage ϕSa2 carrying the PVL genes, the arginine catabolic mobile element (ACME) type I, and the acquisition of mutations causing resistance to fluoroquinolone [43,44]. USA300 has been reported to be independently associated with metastatic complications in patients with bacteremia but not in solid organ transplant recipients [5,45]. However, USA300 in these studies was not defined by the distinctive PFGE pattern but by the following criteria: *S. aureus* belonging to spa-CC008, SCC*mec* type IVa, and the presence of both PVL genes and ACME. One study analyzed the association of USA300 strains and outcome in 50 adult patients with OM. USA300 lineage was defined by PFGE and only 44% had PVL genes. Neither PVL genes nor USA300 strain type were associated with treatment failure [35]. In children, in cases of OM, the specific clone USA300-0114 was associated with bone fractures and the USA300 lineage was associated with both intensive care unit hospitalization and a more severe biological inflammatory syndrome [9,10,14].

In Europe, MSSA CC398 was associated with mortality in patients with bacteremia in a single center study [4]. In two studies, while CC398 was associated with severe infection in DFI, failure of treatment seems to be less common in patients with BJI [25,26]. However, the lack of CC typing of other *S. aureus* in the first study [25] and the small number of patients and different BJI included in the second study [26] make the results difficult to interpret. ST239-MRSA is the dominant MRSA strain in some countries and has been reported to be multidrug resistant and able to produce several toxins, mainly, staphylococcal enterotoxins (SEA, SEK, and SEQ), exfoliative toxins A and B (ETA and ETB), and PVL [46]. In a study in Taiwan from 2016 to 2019, ST239-MRSA has been reported to be associated with elevated inflammatory serum markers and also with stronger biofilm formation than other types (mainly ST8 and ST59) of MRSA in PJI, suggesting that this strain type may be more virulent than others [27].

#### 3.5.2. Role of the *agr* System

The accessory gene regulator (*agr*) locus of *S. aureus* is a quorum-sensing virulence regulator that coordinates the expression of various virulence factors [47]. *agr* system activation leads to decreased production of cell-wall-associated factors, causing the dispersion of biofilm, the spread of an infection, and a simultaneous increase in exoprotein gene expression [48]. Loss of *agr* activity can result in abundant biofilm formation and strains deficient in autolysis, which can contribute to the persistence of infection and poor outcomes [36,49]. *agr* dysfunction is detected phenotypically by an absence of δ-hemolysin activity or absence of δ-toxin production and not by genetic background [50,51]. The effect of *agr* dysfunction in patients with BJI was also not reported in this review. However, *agr* type III was associated with orthopedic complications and progression to chronic OM in children, whereas *agr* type II was associated with resolved infection in adults with PJI [21,29]. This finding could be an effect of clonal origin or genetic differences in toxin prevalence, but no effect of PVL gene carriage or specific clonal complex were associated with outcomes in these studies. Notably in children, *agr* type III was associated in multivariate analysis with orthopedic complications, whereas USA300 strains and strains carrying PVL genes were not statistically different from other strain types in clinical outcomes [21].

#### 3.5.3. Role of PVL Genes

PVL is a toxin composed of two components, LukS-PV and LukF-PV, that, after assembling, forms pores in leukocyte membranes and leads to neutrophil lysis [52]. *S. aureus* carrying PVL genes are more frequent in skin and soft tissue infections than in other more invasive infections [53]. Moreover, PVL is associated with increased infection severity in children and adults with pneumonia and skin and soft tissue infections [54,55,56]. However, the role of PVL in other invasive infections is not clear. In patients with bacteremia, results of studies are contradictory. Studies suggested an association between PVL and prolonged duration of bacteremia, prolonged duration of fever, and higher risk of development of sites of metastatic infection but not with sepsis/septic shock or mortality [57,58,59,60]. In a recent multicenter study in Australia and New Zealand in children with *S. aureus* bacteremia, PVL was the only virulence factor associated with poor outcomes with a composite score comprising 90-day all-cause mortality, 90-day relapse, ICU admission, or length of hospital stay > 30 days (OR [95% CI] = 2.57 [1.04–6.22], *p* = 0.038), but not with 90-day all-cause mortality alone (OR [95% CI] = 1.27 [0.09–12.2], *p* = 0.84) [61]. Given the association of PVL with some severe infections, it would seem possible that the presence of the toxin may impact the clinical outcomes of BJI.

Our review suggests that children with PVL-positive S*. aureus* OM had higher serum inflammatory markers at presentation, a longer course of illness, more days of fever, and more complicated infection (muscle abscess, pyomyositis, subperiosteal abscess, visceral abscess, and deep venous thrombosis). Moreover, blood cultures were more likely to be positive, and children were more likely to require ICU admission [15]. Such a clinical picture should prompt the clinician to consider the possibility of infection by a PVL-positive *S. aureus* strain and to be aware of associated complications such as deep venous thrombosis and multifocal infection.

However, in adults, PVL-positive *S. aureus* was not associated with severity or poor outcomes in BJI. Only one study found an association with risk of chronic OM [24]. This difference may be explained by the fact that in children BJI is often caused by hematogenous spread of bacteria (hematogenous OM), whereas BJI in adult studies are usually more heterogenous. Indeed, the prevalence of PVL is more common in *S. aureus* invasive disease such as severe sepsis and in community-acquired infection (especially for MRSA strains) [43]. Another explanation could be the higher prevalence of PVL-positive *S. aureus* in children with OM compared to adults (median 40% [4.8–100] vs. 7% [0–57]). Differences in the prevalence of PVL-positive *S. aureus* have already been described in a French multicenter study in patients with *S. aureus* pneumonia hospitalized in ICU between 2011 and 2016. In this study, 95% (*n =* 19/20) of patients less than 3 years of age, and 86% (*n =* 6/7) between 3 and 18 years of age had PVL-positive *S. aureus*, compared with only 44% (*n =* 60/136) in older patients [56].

#### 3.5.4. Role of Other Virulence Genes

*S. aureus* displays a vast array of virulence factors, many of which exhibit broad functionality that allow bacteria to adapt to different hosts and environments. *S. aureus* expresses many proteins on its surface. The most well-studied are cell wall proteins belonging to the family of microbial surface components recognizing adhesive matrix molecules (MSCRAMM), which play a key role in *S. aureus* BJI by initiating staphylococcal attachment. MSCRAMM includes clumping factor proteins (ClfA and ClfB) and surface-anchored proteins (SdrC, SdrD, SdrE) [62].

*S. aureu*s biofilms are complex structures produced by bacteria that attach to surfaces and contain communities of *S. aureus*. Biofilms harbor an extensive exoproteome, including proteins with functions related to immune evasion and pathogenesis such as hemolysins, nucleases, lipases, proteases, and collagenases. Among these factors, capsular polysaccharides of serotypes 5 (cap5) and 8 (cap8) are recognized as important virulence factors [6,7]. Despite the main role of these proteins in biofilm formation, few studies in our review reported a higher prevalence of genes encoding these proteins in patients with BJI having poor outcomes. One study reported more than 40 genes significantly associated with severity of illness in children with OM, including enterotoxin genes, capsule gene (*capE)*, PVL genes, and MSCRAMMs such as *cna* and *bbp* [13]. One study in adults with PJI found that *cap5* was associated with treatment failure and *cap8* with treatment success and the eradication of *S. aureus*. Some studies have shown that the loss of capsular polysaccharide production was associated with the persistence of *S. aureus* in humans with chronic OM, whether the *cap5* or *cap8* genes were initially present [63,64]. Additionally, it is important to note that USA300, often thought to be a virulent strain in BJI, does not form a capsule, suggesting that capsule production is not necessary for virulence [65]. These findings highlight the difficulty of analyzing only the presence of genes without correlation using phenotyping studies.

#### 3.5.5. Strengths and Weaknesses of the Literature and Perspectives for the Future

The current review provides the most comprehensive overview to date of the role of *S. aureus* genotyping associated with the outcomes of *S. aureus* BJI. Studies included in this review were identified from a search of the Pubmed database using a systematic search strategy. However, despite having applied stringent inclusion criteria, it is possible that some relevant articles were excluded. It is important to note that treatment as a covariate affecting reported outcomes was not typically reported in this review, mainly due to lack of data. Indeed, treatment could significantly impact the outcomes of BJI regardless of the genetic background of *S. aureus*.

Current literature has certain weaknesses, which could be improved on in future studies.

First, few studies recorded and analyzed important independent factors associated with outcomes. Indeed, several factors play an important role in the failure of treatment in BJI and, specifically, in PJI or in device-associated OM. Patients’ comorbidities, antimicrobial therapy, and surgical treatment are essential to the success of therapy. Future studies should take into account these clinical factors to assess outcomes in patients with BJI, especially in PJI and OM with a foreign body.

Second, the definition of the outcomes varied by study and types of BJI examined, making it difficult to compare results across studies. A homogenous cohort of patients with BJI (e.g., hematogenous PJI or acute PJI, OM with or without device infection) and validated criteria of outcomes, with a minimum duration of follow-up of 1 year, would provide results that could be compared across studies with a clinically relevant outcome [66].

Third, the phenotypic characteristics of *S. aureus* strains, including biofilm and SCV formation are important factors in the poor outcomes of BJI. Studies in this review included a wide range of *S. aureus* genes and virulence factors, but it is important to note that the presence of a given gene does not necessarily imply a specific protein product or cell function. Moreover, the large number of genes analyzed in studies using WGS raises the possibility that some of the associations observed resulted from chance alone. Studies assessing the phenotype and genotype of *S. aureus* are scarce and, in our review, we included mostly studies with a small number of participants, which limited the generalizability of the results. Future studies should include more patients and have a multicenter design to produce reliable results that are generalizable to other populations. Moreover, phenotypic studies are necessary to better understand the role of these genes in the outcomes of BJI. For example, the ability of strains to produce biofilm and its relation to the outcomes of BJI should be addressed along with genetic analyses.

## 4. Conclusions

Whereas PVL genes were associated with poor outcomes in children, no specific genes were reported similarly in adults. Further phenotypic and transcriptomic studies are needed to achieve a better understanding of the influence of virulence factors of *S. aureus* on the evolution and outcomes of BJI.

## Figures and Tables

**Figure 1 ijms-24-03234-f001:**
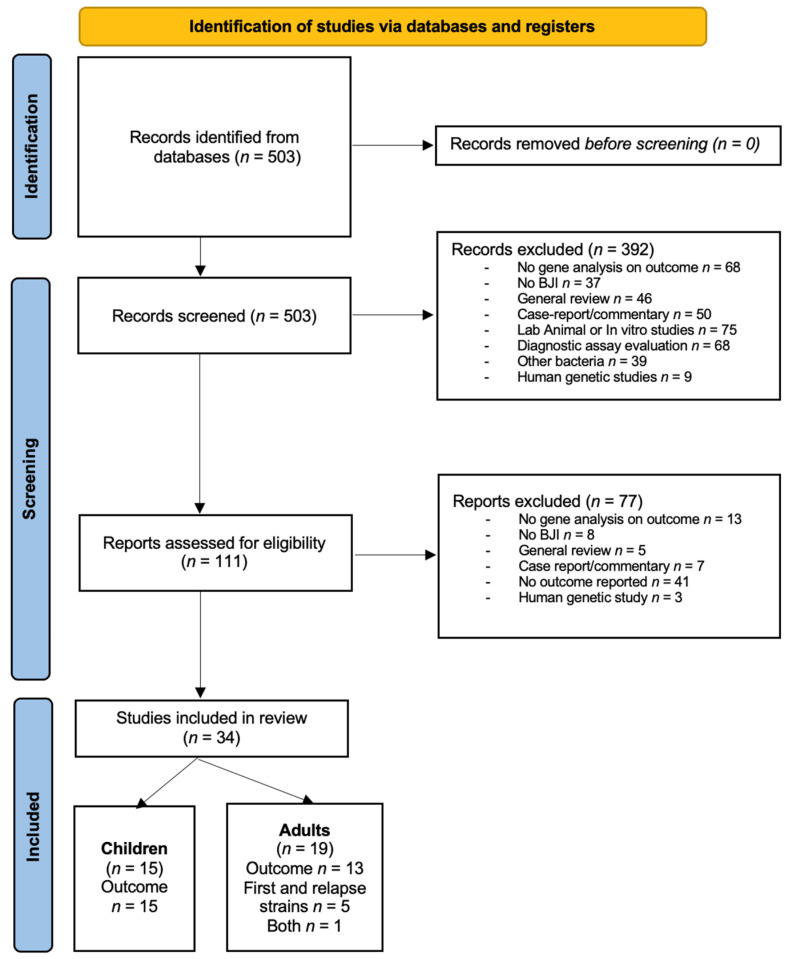
Flow chart of included studies.

**Table 1 ijms-24-03234-t001:** Association between genes and outcome in bone and joint infections in children.

Reference	Study Design	Country	Years of Study	Number of Patients/Strains	MRSA n, (%)	Type of BJI	Genomic Analysis	Groups Compared	Genotype and Outcome
Belthur et al., 2012 [14]	Retrospective, single-center	USA	2001–2009	66/66	58/66 (88)	AHO	PFGE, PCR PVL gene (unspecified)	Bone fracture (*n =* 17) vs. no bone fracture (*n =* 49)	Prevalence of PVL and USA300 clone did not differ significantly PVL: *n =* 17/17, 100% vs. *n =* 45/49, 92% (*p* = 0.565) USA300 pulsotype: *n =* 16/17, 94% vs. *n =* 43/49, 88% (*p* = 0.667) The USA300-0114 clone was more frequent with a fracture: *n =* 14/17, 82% vs. *n =* 27/49, 55% (*p* = 0.034)
Bocchini et al., 2006 [15]	Retrospective, single-center	USA	2001–2004	89/89	56/89 (85)	AHO	PCR *luk-S-PV*, *luk-F-PV*, *fnbB*	PVL+ (*n =* 59) vs. PVL- (*n =* 26) *fnbB*+ (*n =* 66) vs. *fnbB*- (*n =* 10)	PVL+/PVL- Severe infection that required care in the ICU: *n =* 11/59, 18.6% vs. *n =* 1/26, 3.8% (*p* = 0.01) Higher CRP levels (mg/L, mean ± SD): 23.1 ± 18.1 vs. 7.1 ± 7.1 (*p* = 0.000002) Higher WBC count (cells x 10^3^/mm^3^, mean ± SD): 15.1 ± 6.6 vs. 11.6 ± 6.4 (p = 0.03) Positive blood culture: *n =* 39/58, 67.2% vs. *n =* 5/26, 19.2% (*p* = 0.0001) Surrounding myositis/pyomyositis on MRI: *n =* 28/45, 62.2% vs. *n =* 6/19, 31.6% (*p* = 0.05) Subperiosteal/intraosseal abscess on MRI: *n =* 34/45, 75.6% vs. *n =* 9/19, 47.4% (*p* = 0.06) *fnbB*+/*fnbB*− No significant differences
Bouras et al., 2018 [16]	Retrospective, two centers	Greece	2007–2015	125/68	26/68 (38.2)	OM (*n =* 72) septic arthritis (*n =* 45) Both (*n =* 6)	PCR *luk-S-PV*, *luk-F-PV*, *cna*, *fib*, *clfA*, *clfB*, *fnBPA*, *fnBPB*, *bbp*, *eno* and *ebpS* PCR *agr* and SCC*mec*	Presence of virulence genes or not	No differences except for a correlation of PVL positivity with greater number of days of hospitalization (mean ± SD) (PVL+ vs. PVL*−*: 19.94 ± 7.61 vs. 14.47 ± 5.11 days [*p* = 0.002])
Carrillo-Marquez et al., 2009 [17]	Retrospective, single-center	USA	2001–2008	44/44	17/44 (38)	Septic arthritis	PCR *lukS-PV* and *lukFPV*	PVL+ (*n =* 27) vs. PVL− (*n =* 17)	More associated cellulitis/abscess with PVL+: *n =* 8/27, 30% vs. 0% (*p* = 0.02) Initial CRP > 10 mg/L: *n =* 8/22, 36% vs. *n =* 1/16, 7% (*p* = 0.05)
Collins et al., 2018 [13]	Retrospective, single-center	USA	2009–2014	70/70	43/70 (61)	AHO	WGS (201 virulence genes tested)	Severe ^1^ (*n =* 36) vs. non-severe (*n =* 33)	40 genes were significantly associated with increased severity of illness of the affected children (*p* < 0.01) Enterotoxins: *seg*, *sek*, *sei*, *seq*, *sem*, *sen*, *seo* Exotoxins: *set13*, *set11*, *ssl11nm*, *set7*, *set14*, *ssl6nm*, *set15*, *set12* Adhesin: *cna* Hemolysin: *hla* Toxins: *lukS-PV*, *lukF-PV*, *lukE* Proteases*: splA*, *splB* Others: *capE*, *sarU*, *ccra*, *lpl1*, *agrC*, *epiC*, *ear*, *lpl4*, *hsdM*, *sarH2*, *pls*, *cadA*, *spa*, *bbp*, *speG*, *lpl11*, *fhuD*, *epiB* MRSA isolates encoded a significantly greater number of virulence genes (*p* < 0.0001) than did MSSA isolates
Dohin et al., 2007 [18]	Retrospective, single-center	France	ND	31/31	2/31 (6.5)	AHO (*n =* 18) Septic arthritis (*n =* 11) Both (*n =* 2)	PCR *lukS-PV-lukF-PV*	PVL+ (*n =* 14) vs. PVL− (*n =* 17)	More severe sepsis or septic shock with PVL+: *n =* 6/14, 43% vs. *n =* 0/17 (*p* = 0.004) More days of fever with PVL+ (mean ± SD): 28.6 ± 23.2 vs. 2.8 ± 2.2 (*p* < 0.001) More complications with PVL+: Muscle abscess: *n =* 3/14, 21% vs. 0/17 (*p* = 0.08) Pyomyositis: *n =* 5/14, 36% vs. 0/17 (*p* = 0.01) Subperiosteal abscess: *n =* 11/14, 79% vs. *n =* 1/17, 6% (*p* < 0.001) Visceral abscess: *n =* 11/14, 79% vs. 0/17 (*p* = 0.012) Deep venous thrombosis: *n =* 3/14, 21% vs. *n =* 0/17 (*p* = 0.08)
Gaviria-Agudelo et al., 2015 [12]	Prospective, single-center	USA	ND	12/12	12/12 (100)	AHO	WGS (68 virulence genes tested)	Severe ^1^ (*n =* 8) vs. non-severe (*n =* 4)	Univariate analysis did not identify any virulence gene or SNP associated with severity *sek* + *ear* genes combination were more frequent in severe infection: *n =* 4/8, 50% vs. *n =* 0/4 (*p* = 0.208)
Gijon et al., 2020 [19]	Prospective, multicenter	Europe	2012–2014	85/85	5/85 (6)	OM (*n =* 62) Septic arthritis (*n =* 21) Pyomyositis (*n =* 2)	PCR *lukS-PV lukF-PV*, *mecA*	Severe (ICU or death) (*n =* 11) vs. non-severe (*n =* 74)	The multivariate analysis identified PVL genes carriage as the only factor independently associated with severe outcomes aOR (95% CI): 12.5 (1.75–89) (*p* = 0.01)
Kok et al., 2018 [9]	Retrospective, single-center	USA	2011–2016	183/183	0	AHO and septic arthritis	PCR *lukS-PV*, *lukF-PV* and *agr*	ICU (*n* = 11) vs. Non-ICU (*n =* 172)	In univariate analysis, ICU admission was more often associated with: Bone abscesses: *n =* 7/11, 63.6% vs. *n =* 67/172, 39% (*p* = 0.1) USA300 pulsotype: *n =* 6/11, 54.5% vs. *n =* 42/172, 24.4% (*p* = 0.04) PVL-positive: *n =* 7/11, 63.6% vs. *n =* 36/172, 20.9% (*p* = 0.004) Vancomycin MIC ≥ 2 ug/mL: *n =* 2/11, 18.1% vs. *n =* 5/172, 2.9% (*p* = 0.059) In multivariable analysis, only vancomycin MIC was significantly associated with ICU admission (*p* = 0.049; OR [95% CI] = 7.85 [1.01–61.6])
Martinez-Aguilar et al., 2004 [20]	Retrospective, single-center	USA	2000–2002	59/59	31/59 (52.5)	OM (*n =* 48) Septic arthritis (*n =* 6) Pyomyositis (*n =* 5)	PCR *clfA*, *cna*, *fnbB*, *mecA*, *lukS-PV*, *lukF-PV* and *tst*	PVL+ (*n =* 33) vs. PVL− (*n =* 23)	More complicated infection ^2^: *n =* 10/33, 30% vs. *n =* 0/23 (*p* = 0.002) More febrile days (mean±SD): 4.2 ± 3.6 vs. 2.1 ± 2.5 (*p* = 0.017) No other virulence genes were associated with complications
McCaskill et al., 2007 [10]	Prospective, single-center	USA	2001–2006	72/72	0	OM	PFGE, PCR PVL genes (unspecified genes)	USA300 (*n =* 20) vs. Others (*n =* 52) PVL+ (*n =* 18) vs. PVL− (*n =* 54)	Patients with USA300 isolates infection had greater ESR values at presentation (*p* = 0.03), and maximum ESR values (*p* = 0.01) than the children with non-USA300 isolates Children infected with a PVL+ isolate had significantly increased values for ESR (*p* = 0.005) and absolute neutrophil count (*p* = 0.04) at presentation and maximum ESR (*p* = 0.002) and CRP (*p* = 0.04) compared to infection with a PVL− isolate
McNeil et al., 2019 [21]	Retrospective, single-center	USA	2011–2017	286/286	79/286 (27.6)	AHO (*n =* 160) Septic arthritis (*n =* 96) Both (*n =* 30)	PCR *lukS-PV*, *lukF-PV* and *agr*	Orthopedic complications ^3^ (*n =* 27) vs. no orthopedic complications (*n =* 259)	Univariate analysis: PVL+: *n =* 14/27, 51.8% vs. *n =* 101/259, 38.9% (*p* = 0.21) and USA300 pulsotype: *n =* 13/27, 48.1% vs. *n =* 113/259, 43.6% (*p* = 0.69) were not significantly different and were not included in the multivariable analysis Multivariable logistic regression model for orthopedic complications: *agr* III: *n =* 6/27, 22.2% vs. *n =* 23/259, 8.8% (*p* = 0.007, OR [95% CI] = 5.05 [1.56–16.31]) Prolonged fever: *n =* 13/27, 48.1% vs. *n =* 58/259, 22.4% (*p* = 0.04, OR [95% CI] = 1.9 [1.79–5.51]) Delayed source control: *n =* 7/27, 25.9% vs. *n =* 21/259, 6.5% (*p* = 0.002, OR [95% CI] = 5.91 [1.95–17.88])
McNeil et al., 2020 [11]	Retrospective, two centers	USA	2011–2018	250/250	0	Isolated AHO (*n =* 162) AHO and septic arthritis (*n =* 88)	PCR *lukS-PV*, *lukF-PV* and *agr* (*n =* 250)	Progression to chronic ^4^ OM (*n =* 13) vs. no progression to chronic OM (*n =* 237)	In multivariable analyses, associations between progression to chronic osteomyelitis and: Multiple surgical debridement *n =* 7/13, 53.8% vs. *n =* 38/237, 15.8% (*p* = 0.01, aOR [95% CI] = 6.99 [1.58–31.1]) Delayed source control *n =* 4/13, 30.7% vs. *n =* 22/237, 9.2% (*p* = 0.03, aOR [95% CI] = 2.59 [1.16–11.11]) *agr*III *n =* 4/13, 30.7% vs. *n =* 36/237, 15.2% (*p* = 0.04, aOR [95% CI] = 1.71 [1.03–7.81]) CzIE *n =* 5/13, 38.5% vs. *n =* 31/237, 13.1% (*p* = 0.03, aOR [95% CI] = 13.4 [1.1–18.21]) When CzIE alone was substituted with CzIE and 1GC treatment, no association with chronic osteomyelitis (*p* = 0.3) PVL was not associated with chronic infection in univariate analysis *n =* 5/13, 38.5% vs. *n =* 47/237, 19.8% (*p* = 0.15)
Moutaouakkil et al., 2022 [22]	Prospective, single-center	Morocco	2017–2018	37/37	1/37 (2.7)	septic arthritis (*n =* 14) OM (*n =* 16) Multifocal abscess (*n =* 7)	Multiplex PCR assay (*luk-S/F-PV* and *mecA*)	PVL+ (*n =* 17) vs. PVL- (*n =* 20)	No significant difference on laboratory data: CRP on admission (mg/L, mean ± SD): 170.2 ± 129.9 vs. 164.5 ± 112.8 (*p* = 0.749) Abnormal X-ray was significantly associated with PVL+ patients: *n =* 6/17, 35.3% vs. *n =* 1/20, 5% (*p* = 0.029) An abnormal ultrasound was not associated with PVL+: *n =* 14/17, 82.4% vs. *n =* 11/20, 57.9% (*p* = 0.219) Repeated surgical procedure was not associated with PVL+: *n =* 6/17, 35.3% vs. *n =* 3/20, 15.8% (*p* = 0.177)
Sdougkos et al., 2007 [23]	Retrospective, two centers	Greece	2005–2006	19/19	5/19 (26)	AHO	PCR *lukS-PV* and *lukF-PV*	PVL+ (*n =* 7) vs. PVL− (*n =* 12)	Maximum ESR (mean ± SD): 108 ± 16 vs. 70 ± 15 (*p* = 0.0011) Days until ESR ormalization (mean ± SD): 27.0 ± 3.9 vs. 17.3±6.9 (*p* = 0.006) Maximal CRP, mg⁄dL (mean ± SD): 18.3 ± 8.0 vs. 7.9 ± 4.9 (*p* = 0.0023) Days until CRP normalization (mean ± SD): 17.1 ± 7.9 vs. 8.3 ± 3.7 (*p* = 0.0078)

1 GC: First generation cephalosporin; *agr*: accessory gene regulator system; AHO: acute hematogenous osteomyelitis; aOR: adjusted odds ratio; BJI: bone and joint infection; CI: confidence interval; CRP: C-reactive protein; CzIE: Cefazolin inoculum effect defined as a high-inoculum cefazolin MIC of ≥16 ng/ml; ESR: erythrocyte sedimentation rate; ICU: intensive care unit; MIC: minimum inhibitory concentration; MRI: magnetic resonance imaging; MRSA: methicillin-resistant *Staphylococcus aureus*; MSSA: methicillin-susceptible *Staphylococcus aureus*; ND: no data; OM: osteomyelitis; PCR: polymerase chain reaction; PFGE: pulsed-field gel electrophoresis; PVL: Panton Valentine leucocidin; SCC*mec*: staphylococcal chromosome cassette *mec*; SNP: single nucleotide polymorphism; WBC: white blood cell count; WGS: whole genome sequencing. ^1^ Modified severity of illness scoring system >5. ^2^ Chronic OM (based on imaging studies and/or histopathology) and/or deep venous thrombosis and/or fracture at the site of the infection. ^3^ Orthopedic complications included chronic OM, pathologic fracture, growth arrest/limb length discrepancy, avascular necrosis, and/or chronic dislocation. ^4^ Chronic OM was defined as the presence of any of the following after receiving ≥ 4 weeks of effective antimicrobial therapy: (a) A sequestrum or permeative lucency in bone visible on plain radiographs, (b) New or worsening physical exam signs or symptoms referable to the infected bone/joint (e.g., pain, swelling, drainage, etc.), or (c) Readmission for the treatment of OM.

**Table 3 ijms-24-03234-t003:** Studies with comparison of strains at index and relapse or during course of therapy.

Reference	Study Design	Country	Years of Study	Population	Number of Patients/Strains	MRSA n, (%)	Type of BJI	Genomic Analysis	Comparison	Interval between First and Second Strain	Genotype and Outcome
Lin et al., 2016 [38]	Case report	Taiwan	2010	Adult	1/5	3/5	Native septic arthritis	WGS	First and persistent (SCV)	6, 10, 37, 41 days	13 genetic differences: deletion of the entire SCC*mec*III, three additional insertions of IS256, 8 point mutations (3 non-coding regions; *fabF*, fatty acid synthase; *mprF*, lysyl-phosphatidylglycerol synthase; *clpX*, ATP-dependent *clp* protease; and two different mutations of *cls2*, cardiolipin synthase); and a single nucleotide insertion (*seq*, enterotoxin). Mutation of the *fabF* gene (encodes a fatty acid synthase) was probably responsible for the SCV phenotype of recurrent strains
Cafiso et al., 2021 [39]	Case report	Italy	ND	Adult	1/2	0	Native knee septic arthritis and tibial osteomyelitis	WGS	First and persistent	20 days	No difference in virulence genes No difference in resistance genotype or phenotype In WGS only one SNP differed between 2 strains
Loss et al., 2019 [40]	Case report	France	ND	Adults	1/2	0	PJI	WGS	First and relapse (SCV)	2 years	A striking difference between the index and relapse-SCV isolates was the loss of a ∼42.5 kb prophage in the genome of the stable relapse-SCV strain (φSa2) Only 5 SNPs identified, of which 2 were intergenic and 3 were intragenic, with only 1 introducing a nonsynonymous mutation Small INDELs (1 bp) inducing truncated proteins, due to a frameshift change in *sasA* (encodes SasA protein, associated with binding of platelets) and in *glpD* (encodes aerobic glycerol-3-phosphate dehydrogenase protein) Deletions leading to the emergence of truncated protein forms for an alpha-beta hydrolase (9 bp INDEL) and a putative serine protease (6 bp INDEL) A 9 bp (3 aa, LysGlyPro488−490) non-truncating deletion within the *rpoB* gene
Ma et al., 2022 [41]	Prospective single-center	USA	ND	Adults	3/13	ND	PJI	WGS	First and relapse (*n =* 1), first and recurrent (*n =* 2)	ND	Mutations in genes coding for proteins involved in fibronectin binding (*ebh*, *fnbA*, *clfA* and *clfB*) systematically distinguished later PJI isolates from the first PJI isolate from each patient Pairwise differences between isolates from the same patient differed by 0–9 SNPs or small insertion-deletion mutations, whereas comparisons from isolates between patients differed by more than 100 mutations Patient 1: *fnb* x3, *clfB* Patient 2: *ebh*, lysostaphin, epimerase, pheromone binding Patient 3: *clf* x2, *ebH* x5, indole 3 pyruvate
Trouillet-Assant et al., 2016 [42]	Retrospective single-center	France	ND	Adults	3/6	0	Osteosynthesis material (*n =* 1) PJI (*n =* 2)	WGS	First (*n =* 3) vs. relapse (*n =* 3)	82, 204, 134 days	The genome sizes with the pairs were nearly identical with the exception of one isolate, in which a plasmid was lost in the relapse isolate. Totals of nine, eight and six variants (SNPs and/or INDELs), respectively, were identified, differentiating the pairs obtained from three patients Only a small number of these variants occurred in coding DNA sequences. No effects on the major regulatory systems that are known to control the expression of virulence in *S. aureus*
Muñoz-Gallego et al., 2022 [33]	Prospective multicenter	Spain	2016–2017	Adults	6/12	2/12 (17)	PJI with DAIR (symptoms < 21 days)	WGS	First (*n =* 3) vs. second samples (relapse [*n =* 3] or persistent [*n =* 3])	144, 78, 25 days for persistent strains, ND for relapse strains	In all cases except one different variants of virulence genes were reported differentiating strain pairs in relapsing and persistent PJIs: *bbp*, *sdrD*, *clfA*, *fnbA* and *fnbB* for 2 patients *fmt* (encodes methionyl-tRNA formyltransferase protein) for one patient Deletion in DNA-3-methyladenine glycosylase in one patient Other mutations in 2 patients

aa: amino acid; BJI: bone and joint infection; bp: base pair; DAIR: debridement, antibiotics, and implant retention; MRSA: methicillin-resistant *Staphylococcus aureus;* ND: no data; PJI: prosthetic joint infection SCV: small colony variant; SNP: single nucleotide polymorphism; WGS: whole genome sequencing.

## Data Availability

Not applicable.

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
