# Peer review of "Staphylococcus aureus Genomic Analysis and Outcomes in Patients with Bone and Joint Infections: A Systematic Review"

_ijms, 2023, doi:10.3390/ijms24043234_

Round 1

Reviewer 1 Report

The authors have produced an excellent summary of the studies of the roles of Staphylococcus aureus genes in S. aureus bone and joint infections. As the authors note, this type of review is challenging due to the lack of any type of standard in studies. In spite of these difficulties the authors have produced a valuable and interesting paper.

The paper is excellent as it is, but I do have a few suggestions:

1. A list of abbreviations at the beginning of the paper would be helpful.

2. In the references your have Staphylococcus aureus written as "Staphylococcus Aureus". It should be italicized and "aureus" should be lower case.

3. In line 306 the names of the genes should be italicized, as they are in the rest of the paper.

Author Response

Dear Editor:

We thank you for the review of our manuscript, entitled “Staphylococcus aureus genomic analysis and outcome in patients with bone and joint infections: a systematic review.”

We are very grateful for the opportunity to address the reviewers’ comments. Below are our responses to their minor comments. We hope that they will meet your expectations, and we thank the reviewer for helping us to improve the quality of the manuscript.

Comment 1. A list of abbreviations at the beginning of the paper would be helpful.

Answer:  We added a list of abbreviations, as suggested.

Comment 2. In the references your have Staphylococcus aureus written as "Staphylococcus Aureus". It should be italicized and "aureus" should be lower case.

Answer:  We have modified references to correct this error.

Comment 3. In line 306 the names of the genes should be italicized, as they are in the rest of the paper.

Answer:  Gene names have been italicized.

Reviewer 2 Report

First of all, I want to thank the editor for the opportunity to review this interesting manuscript.

In general, the manuscript is well-written and the authors compiled a significant amount of evidence. Besides the association of PVL and severity of infection or a worse outcome of BJI in children, a clear association between other genetic features of S. aureus and BJI was not found. Available studies are, however, heterogenous and often lack enough information for direct comparison between studies. The authors point out to shortcomings of reviewed studies and suggest how future studies should be performed to provide more reliable data. The review will be useful for the reader interested in S. aureus pathogenesis or BJI.

I have some minor comments:

Line 26: Prosthesis x prosthetic

Lines 30-38: no reference provided.

Lines 41-43: „few studies“ – only one is referenced. The referenced study by Wang et al 2019, does not suggest a difference in genotype/gene presence between S. aureus from different disease types. Please rephrase the sentence.

Lines 54-55: citations x studies/papers/ publications

Lines 67-69: I suggest merging it with the previous paragraph and starting with: „The following data were extracted….

Lines 147-152: The impact of sequence type/clonal complex in children BJI (lines 148-152) is part of 3.2.3. Other genes. Please separate it as an independent paragraph, to make the children and adult parts correspond to each other.

Line 186: complex clonal x clonal complex

Line 321: S. aureus - Italics

Lines 323 and 331: spa-CC008 x use rather more common clonal designation i.e. ST8/CC8-MRSA-IV

Line 325-326: mecA gene is not a virulence-associated gene, please rephrase the sentence and include also the acquisition of mutation causing resistance to fluoroquinolones.  E.g. Gustave et al ISME Journal 2018 describe nicely the evolution of USA300.

Line 347-349: specify the country and year of study/isolate collection. Replace the „other types“ with STs that were compared.

Line 352: missing word? agr system activation?

Line 356: add reference

Line 358: add reference

Lines 393-395: Please add an explanation of why: 1) the hematogenous origin of OM in children should be associated with either PVL or with severity/poor outcome; 2) why heterogenous origin is not.

Line 425: elaborate x produce/form

Line 429: If I correctly understand there was a lack of data on treatment in most of the studies. So the treatment is not reviewed in the manuscript except few mentions. So I think this should be briefly discussed at the end of the discussion (paragraph 3.5.5) among limitations as the treatment could significantly impact the outcome regardless of the genetic background of S. aureus.

Author Response

Dear Editor:

We thank you for the review of our manuscript, entitled “Staphylococcus aureus genomic analysis and outcome in patients with bone and joint infections: a systematic review.

We are very grateful for the opportunity to address the reviewers’ concerns and comments. Below are our responses to their critiques and questions. We hope that they will meet your expectations, and we thank the reviewer for helping us to improve the quality of the manuscript.

Comment 1 : line 26: Prosthesis x prosthetic

Answer : This correction has been made as suggested.

Comment 2 : lines 30-38: no reference provided.

Answer :  The following references have been added:

1.   Patel, R. Periprosthetic Joint Infection. New England Journal of Medicine 2023, 388, 251–262, doi:10.1056/NEJMra2203477.

Commentary:  Recent review on periprosthetic joint infection with highlight of risk factors and treatment of PJI

2.   Turner, N.A.; Sharma-Kuinkel, B.K.; Maskarinec, S.A.; Eichenberger, E.M.; Shah, P.P.; Carugati, M.; Holland, T.L.; Fowler, V.G. Methicillin-Resistant Staphylococcus Aureus: An Overview of Basic and Clinical Research. Nat Rev Microbiol 2019, 17, 203–218, doi:10.1038/s41579-018-0147-4.

Commentary:  Review of the evolution, resistance and role of MRSA according to the type of infection (there is a specific section dedicated to osteomyelitis and PJI)

3.   Urish, K.L.; Cassat, J.E. Staphylococcus Aureus Osteomyelitis: Bone, Bugs, and Surgery. Infect Immun 2020, 88, e00932-19, doi:10.1128/IAI.00932-19.

Commentary:  Comprehensive review of S. aureus osteomyelitis with a specific section on pathogenesis.

Comment 3: Lines 41-43: “few studies“ – only one is referenced. The referenced study by Wang et al 2019, does not suggest a difference in genotype/gene presence between S. aureus from different disease types. Please rephrase the sentence.

Answer: We agree with the reviewer. We propose deleting this sentence because it is not directly related to the topic of our study and results of prior studies are contradictory.

Comment 4: Lines 54-55: citations x studies/papers/ publications

Answer: We have replaced the word “citations” with the word “studies”.

Comment 5: Lines 67-69: I suggest merging it with the previous paragraph and starting with: “The following data were extracted….

Answer: We agree with the reviewer. Because some repetition was present in the original text, we deleted a part of the sentence and merged it with previous section.

Comment 6: Lines 147-152: The impact of sequence type/clonal complex in children BJI (lines 148-152) is part of 3.2.3. Other genes. Please separate it as an independent paragraph, to make the children and adult parts correspond to each other.

Answer:  We added a new section in the part of the manuscript dedicated to pediatric infection (“Sequence type/clonal complex”) that is similar to the corresponding section on adult patients.

Comment 7: Line 186: complex clonal x clonal complex

Answer:  This correction has been made.

Comment 8: Line 321: S. aureus – Italics

Answer:  S. aureus has been italicized

Comment 9: Lines 323 and 331: spa-CC008 x use rather more common clonal designation i.e. ST8/CC8-MRSA-IV

Answer:  Line 323 has been changed with “spa-CC008” to “ST8/CC8-MRSA-IV” which is more commonly used. However, in line 331, spa-CC008 was the method used in these articles to define the USA300 lineage. Therefore, we did not change this line in agreement with the methods in these articles.

Comment 10: Line 325-326: mecA gene is not a virulence-associated gene, please rephrase the sentence and include also the acquisition of mutation causing resistance to fluoroquinolones.  E.g. Gustave et al ISME Journal 2018 describe nicely the evolution of USA300.

Answer:  We modified the sentence as follows:

“Selection and diffusion of this clone is likely related to the acquisition of various mobile genetic elements, specifically SCCmec type IVa carrying the mecA gene, phage fSa2 carrying the PVL genes, the arginine catabolic mobile element (ACME) type I and the acquisition of mutation causing resistance to fluoroquinolone [44,45].”

We added the reference Gustave et al. ISME journal, 2018, as suggested by the reviewer.

Comment 11: Line 347-349: specify the country and year of study/isolate collection. Replace the „other types“ with STs that were compared.

Answer:  We modified the sentence as follows:

“In a study in Taiwan from 2016 to 2019, ST239-MRSA has been reported to be associated with elevated inflammatory serum markers and also with stronger biofilm formation than other types (mainly ST8 and ST59) of MRSA in PJI, suggesting that this strain type may be more virulent than others.”

Comment 12: Line 352: missing word? agr system activation?

Answer: We replaced “agr system” by “agr system activation”, as suggested.

Comment 13: Line 356: add reference

Answer:   We added the following references:

Lee et al. DOI: 10.1038/s41598-020-77729-0

Valour et al. DOI: 10.1016/j.cmi.2015.01.026

Comment 14: Line 358: add reference

Answer:  We modified the sentence as follows:

agr dysfunction is detected phenotypically by an absence of δ-hemolysin activity or absence of δ-toxin production, and not by genetic background [50,51]. The effect of agr dysfunction in patients with BJI was also not reported in this review.”

We added the following references:

Sakoulas et al. DOI: 10.1128/AAC.46.5.1492-1502.2002

Gagnaire et al. DOI: 10.1371/journal.pone.0040660

Comment 15: lines 393-395: Please add an explanation of why: 1) the hematogenous origin of OM in children should be associated with either PVL or with severity/poor outcome; 2) why heterogenous origin is not.

Answer:  We added the following sentence:

“Indeed, the prevalence of PVL is more common in S. aureus invasive disease such as severe sepsis and in community acquired infection (especially for MRSA strains) [44].”

Comment 16: line 425: elaborate x produce/form

Answer:  We replaced the word “elaborate” by the word “form”, as suggested.

Comment 17: line 429, If I correctly understand there was a lack of data on treatment in most of the studies. So the treatment is not reviewed in the manuscript except few mentions. So I think this should be briefly discussed at the end of the discussion (paragraph 3.5.5) among limitations as the treatment could significantly impact the outcome regardless of the genetic background of S. aureus.

Answer:  We added in the section 3.5.5

“It is important to note that treatment as a covariate affecting reported outcomes was not typically reported in this review, mainly due to lack of data. Indeed, treatment could significantly impact the outcome of BJI regardless of the genetic background of S. aureus.
